ecology, environmental science

digital ecology, ecosystem science, remote sensing, 3D mapping, terrestrial laser scanning, structure-from-motion photogrammetry

**Author for correspondence:**
Tim D'Urban Jackson
e-mail: t.d.jackson@bangor.ac.uk

# Three-dimensional digital mapping of ecosystems: a new era in spatial ecology

Tim D'Urban Jackson[1], Gareth J. Williams[1], Guy Walker-Springett[1] and Andrew J. Davies[1,2]

[1]School of Ocean Sciences, Bangor University, Anglesey LL59 5AB, UK
[2]Department of Biological Sciences, University of Rhode Island, Kingston, RI, USA

 TDJ, 0000-0001-7077-2186; GJW, 0000-0001-7837-1619; GW-S, 0000-0002-4544-7573; AJD, 0000-0002-2087-0885

Ecological processes occur over multiple spatial, temporal and thematic scales in three-dimensional (3D) ecosystems. Characterizing and monitoring change in 3D structure at multiple scales is challenging within the practical constraints of conventional ecological tools. Remote sensing from satellites and crewed aircraft has revolutionized broad-scale spatial ecology, but fine-scale patterns and processes operating at sub-metre resolution have remained understudied over continuous extents. We introduce two high-resolution remote sensing tools for rapid and accurate 3D mapping in ecology—terrestrial laser scanning and structure-from-motion photogrammetry. These technologies are likely to become standard sampling tools for mapping and monitoring 3D ecosystem structure across currently under-sampled scales. We present practical guidance in the use of the tools and address barriers to widespread adoption, including testing the accuracy of structure-from-motion models for ecologists. We aim to highlight a new era in spatial ecology that uses high-resolution remote sensing to interrogate 3D digital ecosystems.

## 1. Introduction

Understanding how ecosystems vary in space and time underpins land- and seascape management, but to be effective, accurate and comprehensive information must be captured across multiple scales. Our knowledge of ecosystems represents decades of observations by ecologists using field equipment like quadrats, to capture biological information, and theodolites or satellite positioning systems (e.g. GPS) to record habitat topography. Direct observation field techniques capture detailed habitat information but are labour and resource intensive, resulting in trade-offs between three types of scale: spatial, temporal and thematic, and their components of resolution and extent [1,2]. For example, an abundance survey of all macro-organisms to species level (high thematic resolution and extent) with sampling at 1 m intervals (high spatial resolution) cannot feasibly cover an extent of 1 km$^2$ (limited spatial extent) or if it does, would take a very long time (limited temporal resolution). The impracticality of conventional methods for spatially or temporally continuous sampling has led to an average difference of 5.6 orders of magnitude between the extent represented and extent actually sampled in ecological studies, necessitating interpolation or extrapolation with the risk of over-using data [3].

Disruptive remote sensing technologies to rapidly record detailed, spatially referenced biological and physical information are now accessible to the field ecologist. These techniques overcome some of the logistical challenges and trade-offs of direct observation field sampling and extend the scales of remote sensing capability. This review considers tools able to capture three-dimensional (3D) ecosystem data at finer scales than can be achieved with more familiar remote sensing from satellites or crewed aircraft. We present an introduction to two of the most powerful and accessible high-resolution 3D

mapping techniques, which hold enormous potential for the rapid collection of ecologically relevant, spatially continuous data at multiple scales: terrestrial laser scanning and structure-from-motion photogrammetry (figure 1). Uptake of these new technologies varies widely across disciplines and user groups, and there is a strong case for their increased adoption in ecology. Our primary audiences are ecologists, environmental managers and other interested parties who have limited or no experience with these high-resolution remote sensing tools. We direct more experienced users to our analysis of the accuracy of structure-from-motion photogrammetry models at scales and contexts relevant to ecological studies, addressing a key barrier to uptake. Our aim is to shed light on powerful and increasingly user-friendly tools, encourage innovative and novel analytical approaches, and highlight the new era of 3D digital spatial ecology.

## 2. Remote sensing in ecology

Remote sensing from satellite and crewed aircraft has revolutionized spatial ecology with diverse applications that continue to grow as technology advances in capability, accessibility and familiarity. Passive earth observation from satellites has enabled global-scale mapping and monitoring of land cover, ecosystem function and climatic variables [4], and now offers metre-resolution daily imagery of anywhere on the globe, presenting new opportunities for ecology, conservation and management [5]. Active spaceborne sensors have facilitated the study of broad-scale (kilometre to global) ecosystem structure [6], enabling the estimation of global ocean bathymetry [7] and continuous global topography [8]. The ICESat-2 laser altimetry mission will have ecosystem characterization applications through mapping heights of ice, vegetation canopy and freshwater bodies [9], as well as the unanticipated potential for near-shore bathymetric mapping [10].

Remote sensing from crewed aircraft provides similar data products to satellite sources at higher resolution over smaller extents. Airborne laser scanning has become a widely used tool for characterizing 3D habitat structural complexity and exploring organism–habitat relationships [11,12]. Bespoke or repeat airborne laser scanning surveys are uncommon in academic research owing to high operating costs of crewed aircraft, and compatibility issues pose challenges for the analysis of existing available data [13].

Satellite and crewed aircraft remote sensing is irreplaceable for continuous mapping at up to global extents. However, the technique becomes logistically inappropriate when detailed information is required across smaller spatial extents (metres to hectares) or shorter time periods (hours to weeks) owing to limits of data resolution, accuracy or cost. For 3D mapping at these scales, recent technological advances have led to the emergence of high-resolution (millimetre to centimetre), rapidly deployable remote sensing tools that include terrestrial laser scanning and structure-from-motion photogrammetry (figure 1) [14–16]. Advancement in sampling technology drives an ever-expanding range of questions we can ask about the natural world, and the ability to accurately map ecosystems in three or more dimensions is changing the way we study their ecology and management [11,13,17].

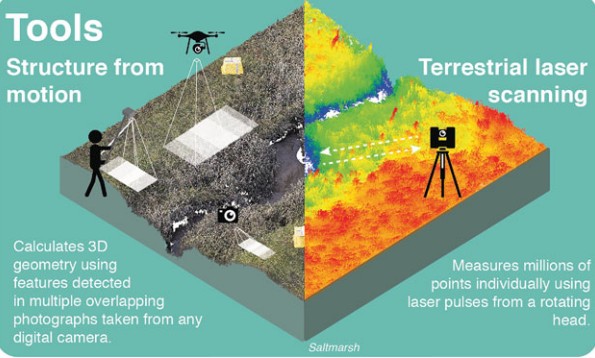

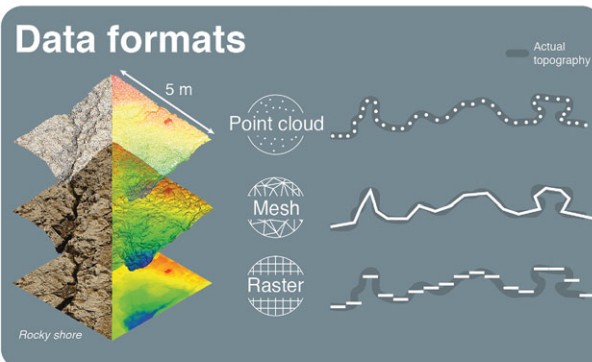

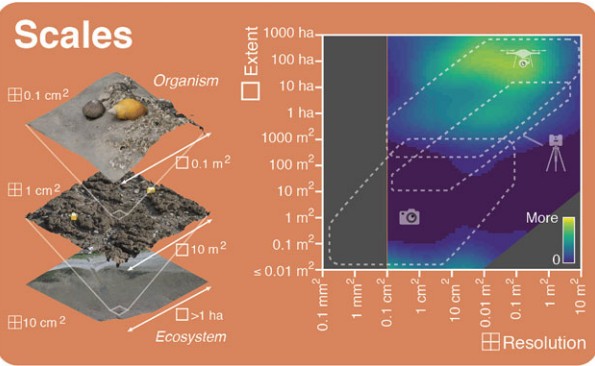

**Figure 1.** An overview of high-resolution three-dimensional (3D) ecosystem mapping tools, data formats and scales. Tools include terrestrial laser scanning and structure-from-motion photogrammetry. Point cloud data can be processed into mesh formats by interpolating between points, and raster formats to produce digital elevation models by averaging point elevations in a regular two-dimensional grid. 3D information can be analysed at multiple spatial scales from organism to ecosystem. These tools enable investigation at spatial scales (resolution and extent) that are understudied in ecology. Plot shading (adapted from [3]) indicates number of ecological studies at specific scales, dashed areas represent the approximate sampling scales for terrestrial laser scanning and structure-from-motion (using drone-mounted and handheld cameras). (Online version in colour.)

## 3. High-resolution remote sensing tools for spatial ecology

Terrestrial laser scanning and structure-from-motion photogrammetry both generate accurate, high-resolution digital 3D models of the environment in the form of a point cloud (figure 1). A point cloud is simply a collection of individual points with $X$, $Y$ and $Z$ coordinates describing their 3D position. Additional attributes can be added to each point to provide information such as colour or other local statistic. From point clouds, other topographical data products like mesh models and rasters can be generated for additional analyses (figure 1). Although their outputs appear similar,

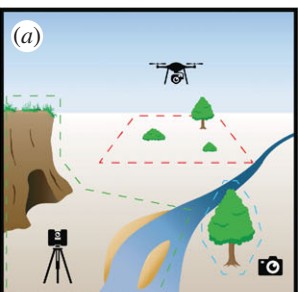 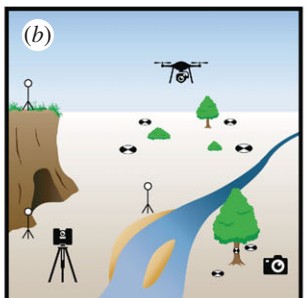 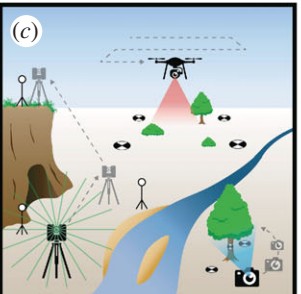 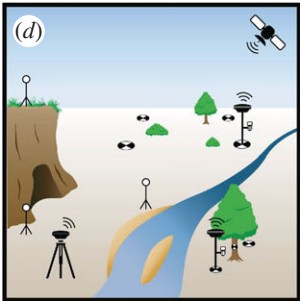

**Figure 2.** Major steps for capturing data with terrestrial laser scanning and structure-from-motion using handheld and drone-mounted cameras. (*a*) Identify features of interest and estimate scanning positions or camera angles. (*b*) Set out reference targets for terrestrial laser scanning, or ground control points, check points and scaling objects for structure-from-motion. For laser scanning, targets are used to align data from different stations, although scene geometry can sometimes be used for alignment instead of, or in addition to targets. For structure-from-motion, reference points are used for aligning images and constraining the modelling process, and for accuracy assessment and scaling. (*c*) Terrestrial laser scanning collects data from a number of discrete stations, to be combined during processing. For structure-from-motion, many overlapping photographs are taken, from which a 3D model is generated during processing. (*d*) Georeferencing, typically using a commercial-grade global navigation satellite system, is required to position the resulting 3D models in real-world space, and for scaling in large structure-from-motion models. (Online version in colour.)

terrestrial laser scanning and structure-from-motion photogrammetry generate point clouds in different ways, resulting in differences in the point cloud characteristics. For an overview of data collection steps using these two techniques see figure 2.

## (a) Terrestrial laser scanning

Using the same principles as airborne laser scanning, terrestrial laser scanning is a high-precision ground-based survey technique used extensively in civil engineering. It is an active remote sensing approach that builds an accurate model of the surroundings by emitting millions of laser pulses in different directions and analysing the reflected signals [18]. Data collected using calibrated laser scanning equipment have intrinsic precision and real-word scale.

Terrestrial laser scanning is conducted from a set of discrete stations using a tripod-mounted instrument, collecting data radially from a low elevation (generally less than 2 m). This results in a reduction in both point density and angle of incidence to the ground with increasing distance from the scanner, and sectors of missing data behind obstructions like trees. Regions with low point density are filled by merging data from multiple scanning stations (figure 2), introducing a low level of quantifiable error. Data extent, resolution and coverage must be balanced with the survey time needed, especially in complex ecosystems like forests where many stations are required for comprehensive coverage of a large extent. Terrestrial laser scanning typically penetrates through fine-scale features like vegetation to record points on internal surfaces (e.g. branches) and the ground, as the independent laser pulses can travel through small gaps. Compared to crewed airborne systems, terrestrial laser scanning offers higher resolution, more accurate data from a near-ground perspective, with lower operating costs and responsive deployment capability, but across a more limited survey extent.

Falling costs and improved portability have increased the accessibility of terrestrial laser scanning to a wide variety of users [15,18]. Custom-built versions have lowered costs even further [19], although the equipment and software required is still expensive compared to structure-from-motion photogrammetry, and may be prohibitively so for some users. Early adoption of terrestrial laser scanning for natural sciences was concentrated in the fields of geography and geoscience [18,20]. More recently it has seen application in ecology [13], particularly in forest ecology where the below-canopy perspective complements airborne data collection. Applications include quantifying biomass, growth and 3D structure of forest vegetation [15,21–24], non-destructive estimation of above-ground grass and mangrove biomass [25,26], assessing vegetation water content [27], studying cave-dwelling bat and bird colonies [28,29], mapping freshwater habitats [30] and exploring the relationships between organisms and fine-scale topography [31,32].

## (b) Structure-from-motion photogrammetry

Structure-from-motion photogrammetry is a low-cost machine vision technique that enables the reconstruction of a detailed 3D model from a set of overlapping two-dimensional (2D) digital photographs [33]. The camera may be handheld or pole-mounted for small scenes, while drone-mounted cameras are commonly used to capture larger extents [34]. Commercial adoption of structure-from-motion is increasing as a low-cost, flexible survey tool, but questions remain over best practices for producing repeatable and high-quality outputs.

With structure-from-motion photogrammetry, the geometry of a scene is reconstructed from the relative positions of thousands of common features detected in multiple photographs taken from different vantages. Structure-from-motion is a passive remote sensing technique because photographs capture reflected light from an external source like the sun. While a basic model can be generated entirely automatically, manual input into the processing stage is required for accurate outputs. Structure-from-motion models have no inherent real-world scale, so known coordinates or distances must be incorporated to generate scale. There is greater opportunity for error introduction with structure-from-motion compared to terrestrial laser scanning, and uncertainty in data outputs varies widely and unpredictably within [35] and among studies [36]. For example, error can be introduced through camera lens distortion, poorly focused images, movement of features in the scene and imprecision in manual processing stages. Care must be taken to minimize the propagation of error through the model construction pipeline [36]. Structure-from-motion generates more homogenous and

comprehensive data coverage compared to terrestrial laser scanning in less time, because the camera is moved around the scene, often using an aerial platform. However, multiple images of a point on a feature are needed to calculate a position, so internal surfaces of complex features (e.g. branches of a dense bush or coral), shaded surfaces and moving features (e.g. blades of grass in the wind) are less likely to be captured or positioned accurately. Structure-from-motion tends to return a generalized outer surface of such features, lacking finer details.

The algorithms used for structure-from-motion are computationally demanding but falling costs of computer processing power and affordable, user-friendly software are making this technique increasingly accessible (see [36] for popular software options). As with terrestrial laser scanning, structure-from-motion saw early adoption in geography and geoscience [33]. Ecological applications include modelling forest and vegetation structure and biomass [25,34,37,38], and quantifying fine-scale habitat topography and structure [14,39–41]. Recently, there has been particular interest in underwater structure-from-motion for measuring and mapping 3D habitat complexity in coral reef systems [42–44].

## (c) Georeferencing

Georeferencing is required to position 3D data generated using terrestrial laser scanning and structure-from-motion in real-world space. Positions of equipment (e.g. laser scanner, drone) or identifiable features (e.g. targets) are typically recorded using a survey-grade global navigation satellite system with an accuracy of 1–3 cm. This stage can represent one of the largest sources of error in the 3D modelling processing pipeline. The influence of georeferencing error on terrestrial laser scanning and small-extent structure-from-motion data (e.g. less than $100 \, \text{m}^2$) can be minimized by incorporating it at a late stage in processing, and with low weighting. However, with large scenes modelled with structure-from-motion using drones, georeferencing using well-distributed ground control points must be incorporated into the process at an earlier stage to provide scale, and prevent warping of geometry [45]. With sub-centimetre-resolution 3D data, georeferencing error can be a limiting factor for the detection of fine-scale change in topography through time [32], and for estimating the accuracy of survey techniques [46], demanding positioning technology with sub-centimetre accuracy (e.g. Total Station).

## 4. Accuracy of structure-from-motion models in ecological settings

Structure-from-motion photogrammetry can achieve impressive accuracy, but the flexibility of the technique makes it vulnerable to the introduction of error that is method and context specific. Most assessments of accuracy in natural settings have been in the field of geoscience, with measurement error varying from less than 1 mm to over 3 m and somewhat dependent on the distance between camera and surface [36]. The spatial scales of ecological patterns often include the very fine (less than 10 cm), so an estimate of the realistic achievable accuracy of structure-from-motion photogrammetry is crucial to assess its usefulness to ecologists and environmental managers.

We compared structure-from-motion and terrestrial laser scanning models within three habitats (rocky shore, honeycomb worm (*Sabellaria alveolata*) biogenic reef and saltmarsh) and at three ecologically relevant scales (fine-scale: $25 \, \text{m}^2$ with less than 1 cm resolution, medium-scale: $2500 \, \text{m}^2$ with less than 2 cm resolution and broad-scale: $2500 \, \text{m}^2$ with 5 cm resolution). Fine-scale photographs were collected using a pole-mounted camera (Canon EOS M, 22 mm lens), while medium- and broad-scale photographs were collected using a drone (DJI Phantom 3 Pro) flying at 25 m and 90 m altitude, respectively. Terrestrial laser scanning data were used as 'truth' because it is a commercially recognized technique with known precision (6 mm at 50 m range), and the most accurate 3D mapping technique we were aware of. Structure-from-motion and terrestrial laser scanning surveys were conducted simultaneously using shared reference targets, to avoid the introduction of georeferencing error. Survey and data processing protocols were designed to achieve maximum accuracy. Models were compared as point clouds using the M3C2 algorithm implemented in the open-source software CLOUDCOMPARE, designed for comparison of 3D point clouds from natural scenes containing surface complexity at multiple scales [47,48]. A comparison of point cloud data avoided the introduction of error by the more common approach of interpolating and averaging data to a raster format digital elevation model [46]. For detailed methods see the electronic supplementary material, S1.

We found mean absolute distance (±1 s.d.) between structure-from-motion and terrestrial laser scanner data ranged from $4 \pm 14 \, \text{mm}$ (fine-scale, rocky shore) to $56 \pm 111 \, \text{mm}$ (medium-scale, saltmarsh) (figure 3). In all cases, distances between the point clouds clustered close to zero, indicating good average agreement, with positive and negative errors compensating each other. The spread of measured distances varied, with fine-scale and stable substrate scenes showing the least variation, while broad-scale and vegetated scenes showed the most (figure 3). Visual inspection of model difference maps and cross-sections revealed that on average structure-from-motion models were accurate, but as resolution decreased, sharp features became smoothed, with cuboid reference objects being represented as mounds (electronic supplementary material, figure S2). Similar results are reported in other studies, with high agreement between structure-from-motion and terrestrial laser scanning at fine-scales of up to $1 \, \text{m}^2$ [25,49] and centimetre-level accuracy at broad scales (hectares) [46,50].

## 5. A case for increased adoption of three-dimensional mapping techniques in ecology

Terrestrial laser scanning and structure-from-motion photogrammetry offer rapid, detailed, continuous extent 3D mapping of ecosystems. Relieving scale-dependence of sampling and easing trade-offs in scale presents opportunities to ask new questions of the natural world and revisit classical paradigms at new scales. The potential applications for high-resolution 3D mapping techniques are vast, and like satellite remote sensing and airborne laser scanning, much of their value will probably only emerge once techniques are firmly established as standard ecological tools. Unique insights are already being generated, particularly in forest and coral reef ecosystems [51], whereas adoption has been

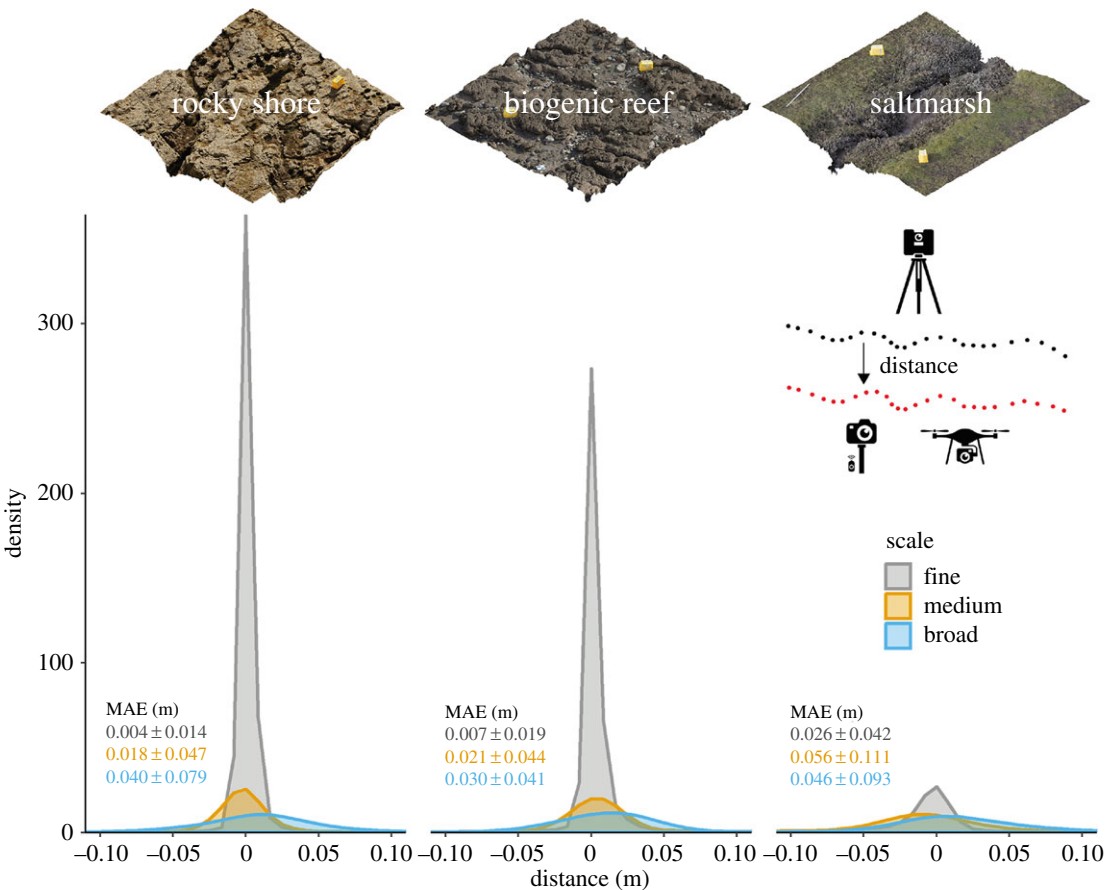

**Figure 3.** Accuracy of a structure-from-motion point cloud quantified as the point-by-point distance to a reference terrestrial laser scanning point cloud in three habitats (rocky shore, biogenic reef and saltmarsh) and at three scales (fine: 25 m$^2$ with less than 1 cm resolution, medium: 2500 m$^2$ with less than 2 cm resolution and broad: 2500 m$^2$ with 5 cm resolution). Distances were measured at 100 000 points and plotted as density curves, with the area under each curve being equal. Curve tails beyond 0 ± 0.1 m are not shown. Mean absolute error (MAE) ± 1 s.d. (m) distance is reported. (Online version in colour.)

slower in other systems such as intertidal habitats. Multiscale topography plays a critical structuring role in the intertidal zone by controlling environmental conditions and field time is constrained by tidal cycles, making rapid 3D mapping tools valuable to intertidal field ecologists. In this section, we identify and discuss several themes of study in which emerging techniques have either already found innovative and transformative applications or are likely to have high impact in the near future (figure 4).

## (a) Understanding relationships between organisms and habitat structure

Analyses of organism–habitat relationships can be hampered by our ability to quantitatively capture the environment. This has resulted in a diversity of definitions, metrics and methods employed to understand the mechanisms behind system-independent phenomena like habitat complexity–biodiversity relationships [52]. The analysis of digital representations of 3D habitat structure to derive system- and scale-independent metrics, like fractal dimension [53], or novel organism-centric metrics [54], could lead to improved understanding by reducing the need to simplify 3D habitat structure (e.g. to 2D profiles) to facilitate analysis [42,43,52,55,56].

Spatial patterning and the patchiness of species across a landscape can depend on topography at multiple scales. In tidal flats and flood plains, elevation changes in the order of centimetres can control species distributions, interactions and ecosystem services [14]. Understanding fine-scale

relationships can improve species distribution and habitat suitability modelling, a valuable management tool, and lead to advances in organism-perspective landscape analysis. Terrestrial laser scanning was used to estimate topographically controlled foraging habitat suitability for the black oystercatcher (*Haematopus bachmani*) and model how it may change under future sea-level rise [31]. Fine-scale topography and 3D structure can control other variables that can be modelled in finer scales than ever before, like microclimate [57], soil pH [58] and hydrodynamic forces [59]. This can enable the quantification of environmental variables as continuous rather than categorical factors, which may lead to alternative or improved interpretations of organism–environment relationships [60,61].

## (b) Measuring and monitoring small, slow and complicated variation in three-dimensional form

Improved morphological descriptions of complex natural shapes can be made with comprehensive 3D data, and variation in such shapes can be monitored through space and time at an organism-relevant resolution. Using terrestrial laser scanning, researchers found that oysters, an ecosystem engineer, can grow reef structure at a faster rate than current sea-level rise, with important management and conservation implications [62]. The coral reef structure is difficult to quantify and previous methods known to poorly capture detailed topography, like the chain-and-tape method, can now be replaced with more repeatable structure-from-motion surveys

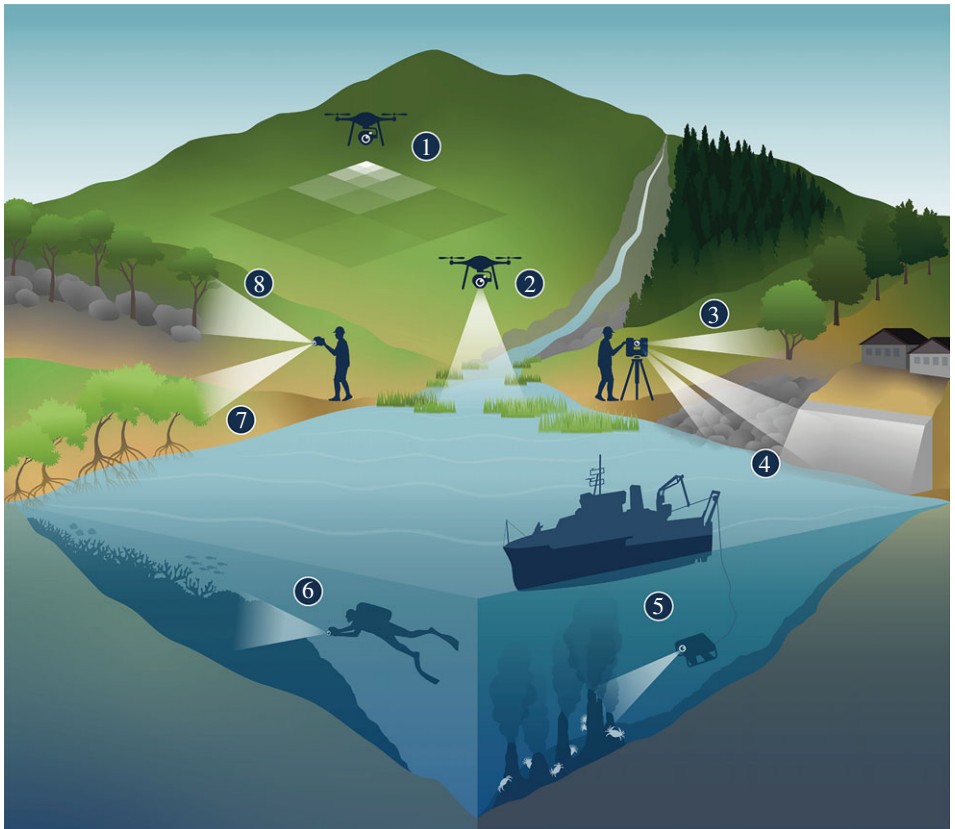

**Figure 4.** Examples in ecology and environmental management with existing or potential applications for 3D ecosystem mapping. (1) Multiscale experimental design with high-resolution 3D mapping across large extents. (2) Mapping fine-scale variation in topography across tidal flats and wetlands. (3) Automated species identification and biometric measurement in forests. (4) Comparing topographic variation in natural and artificial hard coastal substrates. (5) Digital archiving of 3D habitat structure in inaccessible ecosystems. (6) Monitoring variation in reef topography in space and time. (7) Modelling growth in complex 3D organisms like mangrove trees. (8) Mapping 3D structure in habitats with canopy cover and overhangs. (Online version in colour.)

with similar in-water effort [42,43]. Through accurate feature modelling, terrestrial laser scanning can improve on traditional allometric equation methods to estimate above-ground biomass in trees (9.68% overestimation compared to 36.57–29.85% underestimation) [63]. The low cost of operation and rapid deployment capability of terrestrial laser scanning and structure-from-motion make them suitable for opportunistic pre- and post-event change detection [64] and environmental impact assessment monitoring.

## (c) Virtual sampling, digital archiving and addressing problems of scale in ecology

With sampling now achievable at sub-centimetre resolutions, ecosystems can be digitally captured to a degree that in some instances exceeds the resolution possible using *in situ* human observation. There are, however, still limitations of completely removing the human observer element. Macroalgal canopy cover estimates on rocky shores are indistinguishable between 'virtual quadrats' from drone-derived image mosaics and *in situ* human observers using field quadrats, but understory turfing algal species are under-sampled in virtual quadrats [65]. A sampling of cryptic species and multi-layered features will remain challenging to sample using remote sensing. Despite some limitations, the potential advantages of sub-centimetre digital mapping of ecosystems are hugely exciting, including automated species detection and identification using machine learning [66], entire extent sampling that removes interpolation issues when scaling up

from replicate samples [3] and simultaneous biological and environmental sampling [65] (figure 4). Capturing and archiving detailed digital snapshots of ecosystems in a rapidly changing world is likely to prove invaluable for the future, currently unknowable analytical approaches.

Organisms interact with their environment at a range of scales, but understanding scale-dependent patterns and processes is a long-standing challenge in ecology [67,68]. Observation of organisms and their environment is often conducted at spatial, temporal and thematic scales that are human-centric and chosen arbitrarily or logistically, rather than guided by the ecological processes being studied [1,67,68]. Owing to the versatility of high-resolution remote sensing methods like terrestrial laser scanning and structure-from-motion, studies can now be conducted at scales that have previously been underexplored in ecology (figure 1) [3]. One of the difficulties in the multiscale analysis is the time and resource constraints of sampling the same extent at different resolutions [1]. With the ability to rapidly sample large extents at high-resolution, multiscale data can be digitally generated by resampling. We have increasing flexibility to move away from arbitrarily chosen sampling scales and observe ecosystems at ecologically relevant and mechanistic scales.

## (d) Value to managers, policymakers and the public

In a rapidly changing world, tools to efficiently record accurate, detailed snapshots of the environment and monitor ecosystem health are extremely valuable to environmental managers and

policymakers. Policymakers require high-quality environmental information to make evidence-based decisions aimed at limiting environmental impact, conserving ecosystems and maintaining ecosystem services, to the benefit of the public. Often, availability of technology to environmental managers is not limiting, but without practical information on how to efficiently use tools, and analyse and interpret new data types with confidence, there may be a lag in adoption of emerging technologies in favour of more familiar methods, despite their known limitations [69,70]. A benefit of high-resolution 3D mapping technologies for public-facing research groups and environmental bodies is the easily interpreted visual data products generated. Photo-realistic 3D models of ecosystems aid explanation of ecological processes and issues, improving public communication and education through digitally annotated still images, animations or virtual reality systems.

## 6. Barriers to wider uptake in ecology

While some sub-disciplines of ecology are making headway in using high-resolution remote sensing methods to answer questions and test ecological paradigms across scales, in general, the methods remain underused across the discipline. A Web of Science search conducted in December 2019 found that just 1.4% (59 out of 4348) of articles about terrestrial laser scanning or structure-from-motion were categorized as 'ecology' compared to 23.7% (1031) categorized as 'geosciences multidisciplinary'. Further, 67.8% of these articles were published in the last 3 years (2017–2019), highlighting the emerging adoption of these techniques. Here, we identify four perceived barriers to wider uptake in ecology.

Firstly, potential users may be unaware that such techniques exist, so a major aim of this article is to introduce ecologists and environmental managers to two of the most common and powerful techniques in an accessible manner. Second, potential users may be somewhat aware of the techniques discussed, but perceive them to be specialized tools and inaccessible owing to high expertize, cost or time requirements. Technological advances in hardware and user-friendly software mean non-specialists can now be using these techniques in a basic form within a day with a small amount of training or self-learning. Equipment, software and training costs can still be significant, especially for terrestrial laser scanning, with further costs incurred for maintenance and insurance. However, the multidisciplinary applications of the techniques mean many institutions will already have access to suitable equipment and software, or can gain access to shared resources. Structure-from-motion costs can be comparable to many other field techniques, especially if using a handheld camera and open-source software. Practical field time requirements are context dependent. In coastal habitats, we found that terrestrial laser scanning took 15–20 min between stations for a typical medium resolution (10 cm point spacing at 100 m range) survey. Structure-from-motion time requirements ranged from approximately 20 min for a $25 \, m^2$ area surveyed using a pole-mounted camera, to 2 h for a 10 ha area surveyed at 2 cm resolution using a multi-rotor drone (45 m altitude). As a photographic technique, structure-from-motion is slowed or halted in low-light, while terrestrial laser scanning can be conducted in darkness. Processing of terrestrial laser scanning data is rapid (1–2 h)

and can even be conducted on a laptop in the field directly after surveying. Processing a basic structure-from-motion model can be achieved in a similar amount of time, but an accurate, detailed model typically takes a day or more to process depending on processing power and number of images. For a comparison of practical considerations for terrestrial laser scanning and structure-from-motion for geoscience see [71].

A third possible barrier to uptake in ecology is that potential users are aware of 3D mapping tools and understand how they are conducted but do not see value in their use, or are resistant to exploring technology-based alternatives to traditional field methods. Technology is unlikely to ever completely replace a human ecologist in the field for direct observation and interpretation, but can augment data collection and improve efficiency and quantification of specific variables if used appropriately [72]. By separating tasks that require human engagement from those that are more efficiently performed using technology, field time can be optimized [65]. These technologies allow us to test existing ecological concepts at novel scales and inspire new questions that could result in novel paradigms and understanding.

Finally, potential users may be aware of the techniques and understand how they are conducted but are sceptical about the accuracy of the outputs at their spatial scales of interest; this is especially relevant for structure-from-motion photogrammetry. To address this, in this paper we have presented results from an assessment specifically to test the realistic accuracy and characteristics of structure-from-motion models in contexts and at spatial scales relevant to ecologists and environmental managers (figure 3). Our results demonstrate that millimetre to centimetre scale variation in topography can be measured in space and time using high-resolution 3D mapping techniques in the field, making them valuable for numerous ecological applications (figure 4).

The perceived barriers to the adoption of 3D mapping techniques for ecological data collection are now low. However, system-specific challenges remain in survey design, data processing and interpretation. With terrestrial laser scanning in complex environments, line-of-sight obstructions and moving vegetation combined with the spatial characteristics of the point cloud data generates challenges for interpretation and analysis [49,73,74]. While the moving vantage aspect of structure-from-motion data capture means more homogenous data coverage, repeatability of coral reef rugosity measurements were impacted by high habitat complexity, environmental conditions and variation in methods [75]. The use of drone-mounted sensors for field ecology comes with an additional suite of considerations for training, permissions and constantly evolving regulations that govern their safe and legal usage [76]. Data processing still requires manual input at various stages, and automated workflows can be computationally demanding, especially for structure-from-motion. Various algorithms and software packages are being developed for 3D point cloud processing, including open-source projects like CLOUDCOMPARE [48]. After the initial processing stages required to generate a 3D model, further processing and analysis currently requires non-trivial technical skill or novel approaches specific to the task. As 3D methods become more common in ecology, an increase in demand and funding for user-friendly and powerful processing techniques, including packages for open-source platforms like Python and R, can be expected.

## 7. Conclusion

Technology is available and accessible to non-specialist ecologists that enables the detailed mapping of habitats and organisms accurately in 3D. These techniques unlock a wealth of new spatial and temporal ecological questions that were logistically impossible to ask only a few years ago. As with any sampling method, the limitations should be understood as uncertainty may not be readily detected, and there is a need for standardization of protocols. The power of these techniques mean they are rapidly becoming standard and essential tools in various disciplines. By embracing emerging technologies, modern ecologists can overcome long-standing challenges in studying scale-dependent organism–environment relationships. Digital ecosystem analysis and multiscale 3D spatial ecology are continuing to evolve, and high-resolution remote sensing techniques are becoming instrumental as part of the modern spatial ecologist's tool kit.

Ethics. Landowner permission was granted for all fieldwork.

Data accessibility. Data and R code supporting this manuscript are available in a figshare repository: https://figshare.com/s/a623925eb42f4fc32e9a.

Authors' contributions. T.D.J., A.J.D. and G.J.W. conceived and drafted the review. T.D.J. and A.J.D. designed the primary data component and T.D.J. and G.W.-S. conducted the fieldwork. T.D.J. processed and analysed the data. All authors helped revise the manuscript and gave final approval for publication.

Competing interests. Gareth J. Williams is an associate editor for Proceedings of the Royal Society B.

Funding. This research was funded by the SEACAMS2 and Ecostructure projects (part-funded by the European Regional Development Fund through the West Wales and the Valleys Programme 2014–2020 and the Ireland Wales Cooperation Programme 2014–2020)

Acknowledgements. We would like to thank Michael Roberts and Jonathan King for facilitating the study, Adel Heenan and Timothy Whitton for their valuable insights, Steve Rowlands and Steve Carrington for their help with data collection and Amanda Dillon for creative contribution to figure 4. We also thank two anonymous reviewers and the handling editor Prof. Innes Cuthill for their comments that greatly improved the manuscript.

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
