## [Reviewer comments · Proceedings of the Royal Society B: Biological Sciences]

Review History

RSPB-2019-2383.R0 (Original submission)

Review form: Reviewer 1

Recommendation

Accept with minor revision (please list in comments)

Scientific importance: Is the manuscript an original and important contribution to its field?

Good

General interest: Is the paper of sufficient general interest?

Good

Quality of the paper: Is the overall quality of the paper suitable?

Excellent

Is the length of the paper justified?

Yes

Should the paper be seen by a specialist statistical reviewer?

No

Do you have any concerns about statistical analyses in this paper? If so, please specify them explicitly in your report.

No

It is a condition of publication that authors make their supporting data, code and materials available - either as supplementary material or hosted in an external repository. Please rate, if applicable, the supporting data on the following criteria.

Is it accessible?

N/A

Is it clear?

N/A

Is it adequate?

N/A

Do you have any ethical concerns with this paper?

No

Comments to the Author

D'Urban Jackson and co-authors have done a very nice job of outlining how access to new ultra-high resolution 3D data has the potential to change how we study ecosystems in space and time. The review is well written and clearly structured, and the great effort put into the graphics really helps convey the potential of these new technologies. I'm sure that this paper would be of interest to a broad readership. Well done!

I've made some suggestions below that hopefully will help the authors refine their paper even further. As a terrestrial vegetation ecologist, my main reservation about the paper is that some of the content around the role of 3D data from TLS and SfM has already been covered in detail by previous reviews. This is particularly true in the context of forests, where TLS data are already revolutionising how we estimate biomass and study forest structure. In fact the Royal Society's Interface Focus journal recently had an entire special issue on exactly this (see here: <https://royalsocietypublishing.org/doi/full/10.1098/rsfs.2017.0052>). The same is true for SfM

data, that are being increasingly used for biomass mapping (e.g., see here for one of the several reviews published on this in the last couple of years:

<https://link.springer.com/article/10.1007/s40725-019-00094-3>). So I guess in my mind the real novelty of what is being proposed in this review is the application of these technologies in new ecosystems (e.g., intertidal, marine etc.) – although I admit that this may simply reflect the fact that I don't know the literature in these systems anywhere near as well as in the context of forests.

The other thing I think is worth mentioning in this context is that in addition to TLS and SfM from drones, the other emerging 3D technology that would be worth incorporating in this review is UAV LiDAR. The technology has come a long way here and there are numerous groups around the world that are capturing extremely high resolution LiDAR from UAVs across scales that can't be covered using a TLS (just as one example of what is possible see here:

<https://www.sciencedirect.com/science/article/abs/pii/S0034425719303748>). So I think it would be worth incorporating this into the review as well as (i) discuss the pros and cons of these data when compared to SfM data from UAVs (e.g., in terms of cost, accuracy, coverage, information etc.) and (ii) discuss how the two can be fused (e.g., <https://royalsocietypublishing.org/doi/full/10.1098/rsfs.2017.0038>).

Other comments:

L132-4: I'm not sure this statement is true. SfM has been rapidly adopted by a whole range of disciplines in ecology and conservation. Moreover, from a commercial standpoint many

surveying companies actually prefer it to LiDAR as the software for processing the drone imagery has come a long way and the data can be acquired much more cheaply, using smaller drones and with less risk.

L141-2: one big difference between TLS (and LiDAR in general) and SfM that would be worth discussing is that LiDAR allows you to see 'through' vegetation and capture both the canopy and the ground, whereas SfM is limited to the surface. This is actually a big limitation in structurally complex ecosystems like forests.

L195: when you say 'low cost' this is not really true for TLS. A good scanner (i.e., REIGL) costs >100K and is out of reach for most groups. By contrast a Phantom 4 drone is about £1K. This is a big distinction between the two technologies that's worth making in the paper.

L194-202: I would also mention the limitation of scale, particularly for TLS. Covering large area in ultra-high detail remains prohibitive in terms of time, cost and computing.

L204: Arguably airborne LiDAR paved the way for high-resolution habitat suitability mapping. Worth mentioning this and discussing what TLS/SfM can give you that airborne LiDAR can't from this perspective. I would think this is likely to be highly dependent on the organism in question and there is likely to be a trade-off with spatial scale (i.e., airborne LiDAR can cover much larger areas).

L240-2: Good on paper, but in reality automating many of these processes is still a big bottleneck. I'm thinking for instance of the algorithm for segmenting trees from TLS that require a lot of manual tuning and pruning to get right. Worth mentioning.

L250-3: I seem to remember similar results from bird colonies (possibly here: <https://besjournals.onlinelibrary.wiley.com/doi/full/10.1111/2041-210X.12974>)

L330-346: I think an additional barrier you are overlooking here are the substantial costs associated with purchasing equipment, getting specialised training, obtaining permits/licences, insurance (for drones in particular) and running costs (batteries, software, part, maintenance etc). In terms of identifying hurdles to uptake one thing that would actually be really interesting to do (although not necessarily for this paper) would be to survey the research community and ask them what the barriers are (e.g., at the BES annual meeting).

Review form: Reviewer 2

Recommendation

Major revision is needed (please make suggestions in comments)

Scientific importance: Is the manuscript an original and important contribution to its field?

Acceptable

General interest: Is the paper of sufficient general interest?

Good

Quality of the paper: Is the overall quality of the paper suitable?

Good

Is the length of the paper justified?

Yes

Should the paper be seen by a specialist statistical reviewer?

No

Do you have any concerns about statistical analyses in this paper? If so, please specify them explicitly in your report.

No

It is a condition of publication that authors make their supporting data, code and materials available - either as supplementary material or hosted in an external repository. Please rate, if applicable, the supporting data on the following criteria.

Is it accessible?

Yes

Is it clear?

Yes

Is it adequate?

Yes

Do you have any ethical concerns with this paper?

No

Comments to the Author

Three-dimensional digital mapping of ecosystems: a new era in spatial ecology

Tim D'Urban Jackson et al.

General comments:

This is a well-written review and I believe fills a need in the literature. The insights are valuable and in my opinion this paper will further help bridge the gap between remote sensing scientists and ecologists. That said, important work must be done to place this review in context of a large body of work undertaken to understand ecological questions using increasingly high detail 3D ecosystem structure using techniques other than TLS and SfM. In addition, the authors should do more work to review or at least mention/cite specific challenges inherent to interpreting digitally derived 3D data of multi-layer/turbid medium ecosystems such as forests.

Specific comments:

Introduction

Line 41: Regarding the "recently emerged" comment: discrete return airborne scanning lidar capable of characterizing 3-D ecosystem structure at the meter scale was relatively mature 20 years ago. Terrestrial laser scanning research to characterize forest structure for ecological applications stretches back about 15 years. It is important to acknowledge this body of work here in the introductory portion of the paper. As is, the authors jump from mentioning "conventional methods" of ecological sampling to the "recently emerged" TLS and SfM methods. Airborne lidar is a critical stepping stone between the two in characterizing 3D ecosystem structure, and is still the only method available to characterize large swaths of land (i.e. tens to hundreds of hectares) in 3D with sub-meter grain.

Line 43: One can easily argue that TLS is not very accessible or powerful for characterizing forest habitats or topographically complex ecosystems (for two examples) across multiple scales, given the large number of scans and logistically challenging geocorrections it takes to characterize these structurally complex environments without significant data dropouts from occlusion effects. The authors are oversimplifying the details of TLS here. ALS needs to be mentioned here in my opinion.

It occurs to me that the way to help focus the paper without confusing readers familiar with the large body of ecological work published on airborne lidar, would be to explain from the outset that the authors are focused on small extent, fine scale ecological questions (i.e. at the scale of

10⁰-10² square meters) where TLS and SfM are most effective.

Paragraph starting Line 55: Again there is a missing set of techniques that are not mentioned here. Airborne passive remote sensing data has long provided sub-meter information relevant to ecological study. Satellite passive remote sensing data also is highly relevant to high spatial resolution ecological processes (e.g. the Worldview series of sensors has gotten progressively finer in resolution over the past decade, to a point of submeter resolution panchromatic imagery, and just above one meter resolution multispectral imagery). And, as mentioned above airborne lidar has been providing sub-meter data for about 20 years. The authors should acknowledge these data types and make the case why they do not warrant discussion in an ecological context. Readers can be pointed toward recent applicable reviews such as Eitel et al. 2016 (<http://dx.doi.org/10.1016/j.rse.2016.08.018>) and Anderson and Gaston 2013 (<https://doi.org/10.1890/120150>). Nowak et al, 2019 (<https://doi.org/10.2478/eje-2018-0012>) also is a very recent and relevant paper relevant to the SfM portion of the current review.

Line 81: I would suggest using the term airborne laser scanning here, to show that it is essentially the same technology as terrestrial laser scanning, just mounted on a different platform.

Line 84: Please be more specific about the wording here, as it is confusing. Rather than "calculating the positions of millions of reflected laser pulses emitted in all directions", the instrument is only able to calculate the location of objects struck by the laser when a portion of the laser is then reflected in the exact same direction back to the TLS instrument.

Line 96: Yes this is a nice overview. The aforementioned review by Eitel et al. 2016 described many ecological applications of TLS and should be cited here for further reading.

Paragraph starting Line 164: Yes this is a nice comparison, with good results. Thank you. I suggest that as this is a review, the authors overview some previous work done in ecosystems with more complex multi-layer structure where lower strata are often obscured from view of the drone. For example, the recent paper by Hillman et al. 2019 (10.3390/rs11182118) highlights some of the challenges in complex multi-layer ecosystems.

Line 236: Very good to highlight water-based ecological applications. Exciting new areas of research for SfM

Line 240: While data acquisition may be rapid and relatively low-cost, it should be mentioned that the workflows required to process these data (particularly TLS in complex structure ecosystems) can be greatly time consuming and require non-trivial technical skill. (OK, in reading further I see that you cover this later).

Line 245: yes, very good. This is the type of information/comparison that I was looking for earlier in the paper. It should occur earlier in my opinion in order to highlight the specific niche that this paper wishes to fill. However, the great amount of ecological work at increasingly finer scales using non-TLS and non-SfM approaches that has occurred since Turner et al.'s 2003 review warrants mention as well.

Line 254: Very good point that as one wishes to examine increasingly finer spatial scales, the data must be evaluated at these finer scales to assess accuracy. This will continue to be a challenge, yet a nice problem to be able to try to solve now that the data exist at this scale.

Line 317-329: Please do mention and cite here the challenges (albeit many opportunities exist as well_ associated with doing fine-scale ecology using these techniques in more structurally complex ecosystems. Citing work in forests for example should be done to make this paper more broadly applicable as a review.

Line 351: to clarify and help differentiate your review from previous work please mention the fine scales you are focusing on, e.g. "...spatial ecology questions at the sub-square meter to tens-of square meters scale..."

Line 357: please acknowledge previous work by acknowledging your review adds to the continuum of ecological work enabled via remote sensing at increasingly finer scales, e.g. "...spatial ecology is continuing to evolve, and high-resolution..."

Figures: The figures are all relevant to the paper in my opinion, have useful captions, and have beautiful aesthetic appeal.

Decision letter (RSPB-2019-2383.R0)

25-Nov-2019

Dear Dr D'Urban Jackson:

Your manuscript has now been peer reviewed and the reviewers' comments (not including confidential comments to the Editor) are included at the end of this email for your reference. As you will see, the reviewers like the topic and your manuscript, but think some further work needs to be done. The good news is that they provide excellent guidance on what is required, so I would like to invite you to revise your manuscript to address their concerns.

Research ethics:

Use of animals and field studies:

Please submit a copy of your revised paper within three weeks. If we do not hear from you within this time your manuscript will be rejected. If you are unable to meet this deadline please let us know as soon as possible, as we may be able to grant a short extension.

Best wishes,
Innes Cuthill

Prof. Innes Cuthill,
Reviews Editor, Proceedings B.
mailto: proceedingsb@royalsociety.org

Reviewer(s)' Comments to Author:

Referee: 1

Comments to the Author(s)

D'Urban Jackson and co-authors have done a very nice job of outlining how access to new ultra-high resolution 3D data has the potential to change how we study ecosystems in space and time. The review is well written and clearly structured, and the great effort put into the graphics really helps convey the potential of these new technologies. I'm sure that this paper would be of interest to a broad readership. Well done!

I've made some suggestions below that hopefully will help the authors refine their paper even further. As a terrestrial vegetation ecologist, my main reservation about the paper is that some of

the content around the role of 3D data from TLS and SfM has already been covered in detail by previous reviews. This is particularly true in the context of forests, where TLS data are already revolutionising how we estimate biomass and study forest structure. In fact the Royal Society's Interface Focus journal recently had an entire special issue on exactly this (see here: <https://royalsocietypublishing.org/doi/full/10.1098/rsfs.2017.0052>). The same is true for SfM data, that are being increasingly used for biomass mapping (e.g., see here for one of the several reviews published on this in the last couple of years: <https://link.springer.com/article/10.1007/s40725-019-00094-3>). So I guess in my mind the real novelty of what is being proposed in this review is the application of these technologies in new ecosystems (e.g., intertidal, marine etc.) – although I admit that this may simply reflect the fact that I don't know the literature in these systems anywhere near as well as in the context of forests.

The other thing I think is worth mentioning in this context is that in addition to TLS and SfM from drones, the other emerging 3D technology that would be worth incorporating in this review is UAV LiDAR. The technology has come a long way here and there are numerous groups around the world that are capturing extremely high resolution LiDAR from UAVs across scales that can't be covered using a TLS (just as one example of what is possible see here: <https://www.sciencedirect.com/science/article/abs/pii/S0034425719303748>). So I think it would be worth incorporating this into the review as well as (i) discuss the pros and cons of these data when compared to SfM data from UAVs (e.g., in terms of cost, accuracy, coverage, information etc.) and (ii) discuss how the two can be fused (e.g., <https://royalsocietypublishing.org/doi/full/10.1098/rsfs.2017.0038>).

Other comments:

L132-4: I'm not sure this statement is true. SfM has been rapidly adopted by a whole range of disciplines in ecology and conservation. Moreover, from a commercial standpoint many surveying companies actually prefer it to LiDAR as the software for processing the drone imagery has come a long way and the data can be acquired much more cheaply, using smaller drones and with less risk.

L141-2: one big difference between TLS (and LiDAR in general) and SfM that would be worth discussing is that LiDAR allows you to see 'through' vegetation and capture both the canopy and the ground, whereas SfM is limited to the surface. This is actually a big limitation in structurally complex ecosystems like forests.

L195: when you say 'low cost' this is not really true for TLS. A good scanner (i.e., REIGL) costs >100K and is out of reach for most groups. By contrast a Phantom 4 drone is about £1K. This is a big distinction between the two technologies that's worth making in the paper.

L194-202: I would also mention the limitation of scale, particularly for TLS. Covering large area in ultra-high detail remains prohibitive in terms of time, cost and computing.

L204: Arguably airborne LiDAR paved the way for high-resolution habitat suitability mapping. Worth mentioning this and discussing what TLS/SfM can give you that airborne LiDAR can't from this perspective. I would think this is likely to be highly dependent on the organism in question and there is likely to be a trade-off with spatial scale (i.e., airborne LiDAR can cover much larger areas).

L240-2: Good on paper, but in reality automating many of these processes is still a big bottleneck. I'm thinking for instance of the algorithm for segmenting trees from TLS that require a lot of manual tuning and pruning to get right. Worth mentioning.

L250-3: I seem to remember similar results fro bird colonies (possibly here: <https://besjournals.onlinelibrary.wiley.com/doi/full/10.1111/2041-210X.12974>)

L330-346: I think an additional barrier you are overlooking here are the substantial costs

associated with purchasing equipment, getting specialised training, obtaining permits/licences, insurance (for drones in particular) and running costs (batteries, software, part, maintenance etc). In terms of identifying hurdles to uptake one thing that would actually be really interesting to do (although not necessarily for this paper) would be to survey the research community and ask them what the barriers are (e.g., at the BES annual meeting).

Referee: 2

Comments to the Author(s)

Three-dimensional digital mapping of ecosystems: a new era in spatial ecology

Tim D'Urban Jackson et al.

General comments:

This is a well-written review and I believe fills a need in the literature. The insights are valuable and in my opinion this paper will further help bridge the gap between remote sensing scientists and ecologists. That said, important work must be done to place this review in context of a large body of work undertaken to understand ecological questions using increasingly high detail 3D ecosystem structure using techniques other than TLS and SfM. In addition, the authors should do more work to review or at least mention/cite specific challenges inherent to interpreting digitally derived 3D data of multi-layer/turbid medium ecosystems such as forests.

Specific comments:

Introduction

Line 41: Regarding the “recently emerged” comment: discrete return airborne scanning lidar capable of characterizing 3-D ecosystem structure at the meter scale was relatively mature 20 years ago. Terrestrial laser scanning research to characterize forest structure for ecological applications stretches back about 15 years. It is important to acknowledge this body of work here in the introductory portion of the paper. As is, the authors jump from mentioning “conventional methods” of ecological sampling to the “recently emerged” TLS and SfM methods. Airborne lidar is a critical stepping stone between the two in characterizing 3D ecosystem structure, and is still the only method available to characterize large swaths of land (i.e. tens to hundreds of hectares) in 3D with sub-meter grain.

Line 43: One can easily argue that TLS is not very accessible or powerful for characterizing forest habitats or topographically complex ecosystems (for two examples) across multiple scales, given the large number of scans and logistically challenging geocorrections it takes to characterize these structurally complex environments without significant data dropouts from occlusion effects. The authors are oversimplifying the details of TLS here. ALS needs to be mentioned here in my opinion.

It occurs to me that the way to help focus the paper without confusing readers familiar with the large body of ecological work published on airborne lidar, would be to explain from the outset that the authors are focused on small extent, fine scale ecological questions (i.e. at the scale of 10^0 - 10^2 square meters) where TLS and SfM are most effective.

Paragraph starting Line 55: Again there is a missing set of techniques that are not mentioned here. Airborne passive remote sensing data has long provided sub-meter information relevant to ecological study. Satellite passive remote sensing data also is highly relevant to high spatial resolution ecological processes (e.g. the Worldview series of sensors has gotten progressively finer in resolution over the past decade, to a point of submeter resolution panchromatic imagery, and just above one meter resolution multispectral imagery). And, as mentioned above airborne lidar has been providing sub-meter data for about 20 years. The authors should acknowledge these data types and make the case why they do not warrant discussion in an ecological context.

Readers can be pointed toward recent applicable reviews such as Eitel et al. 2016

(<http://dx.doi.org/10.1016/j.rse.2016.08.018>) and Anderson and Gaston 2013

(<https://doi.org/10.1890/120150>). Nowak et al, 2019 (<https://doi.org/10.2478/eje-2018-0012>)

also is a very recent and relevant paper relevant to the SfM portion of the current review.

Line 81: I would suggest using the term airborne laser scanning here, to show that it is essentially the same technology as terrestrial laser scanning, just mounted on a different platform.

Line 84: Please be more specific about the wording here, as it is confusing. Rather than “calculating the positions of millions of reflected laser pulses emitted in all directions”, the instrument is only able to calculate the location of objects struck by the laser when a portion of the laser is then reflected in the exact same direction back to the TLS instrument.

Line 96: Yes this is a nice overview. The aforementioned review by Eitel et al. 2016 described many ecological applications of TLS and should be cited here for further reading.

Paragraph starting Line 164: Yes this is a nice comparison, with good results. Thank you. I suggest that as this is a review, the authors overview some previous work done in ecosystems with more complex multi-layer structure where lower strata are often obscured from view of the drone. For example, the recent paper by Hillman et al. 2019 (10.3390/rs11182118) highlights some of the challenges in complex multi-layer ecosystems.

Line 236: Very good to highlight water-based ecological applications. Exciting new areas of research for SfM

Line 240: While data acquisition may be rapid and relatively low-cost, it should be mentioned that the workflows required to process these data (particularly TLS in complex structure ecosystems) can be greatly time consuming and require non-trivial technical skill. (OK, in reading further I see that you cover this later).

Line 245: yes, very good. This is the type of information/comparison that I was looking for earlier in the paper. It should occur earlier in my opinion in order to highlight the specific niche that this paper wishes to fill. However, the great amount of ecological work at increasingly finer scales using non-TLS and non-SfM approaches that has occurred since Turner et al.'s 2003 review warrants mention as well.

Line 254: Very good point that as one wishes to examine increasingly finer spatial scales, the data must be evaluated at these finer scales to assess accuracy. This will continue to be a challenge, yet a nice problem to be able to try to solve now that the data exist at this scale.

Line 317-329: Please do mention and cite here the challenges (albeit many opportunities exist as well_ associated with doing fine-scale ecology using these techniques in more structurally complex ecosystems. Citing work in forests for example should be done to make this paper more broadly applicable as a review.

Line 351: to clarify and help differentiate your review from previous work please mention the fine scales you are focusing on, e.g. “...spatial ecology questions at the sub-square meter to tens-of square meters scale...”

Line 357: please acknowledge previous work by acknowledging your review adds to the continuum of ecological work enabled via remote sensing at increasingly finer scales, e.g. “...spatial ecology is continuing to evolve, and high-resolution...”

Figures: The figures are all relevant to the paper in my opinion, have useful captions, and have beautiful aesthetic appeal.

Author's Response to Decision Letter for (RSPB-2019-2383.R0)

See Appendix A.

RSPB-2019-2383.R1 (Revision)

Review form: Reviewer 2

Recommendation

Accept as is

Scientific importance: Is the manuscript an original and important contribution to its field?

Good

General interest: Is the paper of sufficient general interest?

Excellent

Quality of the paper: Is the overall quality of the paper suitable?

Good

Is the length of the paper justified?

Yes

Should the paper be seen by a specialist statistical reviewer?

No

Do you have any concerns about statistical analyses in this paper? If so, please specify them explicitly in your report.

No

It is a condition of publication that authors make their supporting data, code and materials available - either as supplementary material or hosted in an external repository. Please rate, if applicable, the supporting data on the following criteria.

Is it accessible?

Yes

Is it clear?

Yes

Is it adequate?

Yes

Do you have any ethical concerns with this paper?

No

Comments to the Author

The authors have done an admirable and thoughtful job of responding to comments by the two reviewers. I believe that this will be a useful contribution to the literature in that it should continue to expand the frontier of ecological applications of remote sensing data. Thank you.

Decision letter (RSPB-2019-2383.R1)

16-Jan-2020

Dear Mr D'Urban Jackson

I am pleased to inform you that your manuscript RSPB-2019-2383.R1 entitled "Three-dimensional digital mapping of ecosystems: a new era in spatial ecology" has been accepted for publication in Proceedings B.

The referee has recommended publication without further revision. Therefore, I invite you to

upload the final version of your manuscript. Because the schedule for publication is very tight, it is a condition of publication that you submit the final version of your manuscript within 7 days. If you do not think you will be able to meet this date please let us know.

To upload your manuscript, log into <https://mc.manuscriptcentral.com/prsb> and enter your Author Centre, where you will find your manuscript title listed under "Manuscripts with Decisions." Under "Actions," click on "Create a Revision." Your manuscript number has been appended to denote a revision. You will be unable to make your revisions on the originally submitted version of the manuscript. Instead, revise your manuscript and upload a new version through your Author Centre.

If you wish to submit your data to Dryad (<http://datadryad.org/>) and have not already done so

you can submit your data via this link

Best wishes,
Innes

Prof Innes Cuthill
Reviews Editor, Proceedings B
mailto: proceedingsb@royalsociety.org

Reviewer(s)' Comments to Author:

Referee: 2

Comments to the Author(s)

The authors have done an admirable and thoughtful job of responding to comments by the two reviewers. I believe that this will be a useful contribution to the literature in that it should continue to expand the frontier of ecological applications of remote sensing data. Thank you.

Decision letter (RSPB-2019-2383.R2)

20-Jan-2020

Dear Mr D'Urban Jackson

I am pleased to inform you that your manuscript entitled "Three-dimensional digital mapping of ecosystems: a new era in spatial ecology" has been accepted for publication in Proceedings B.

If you are likely to be away from e-mail contact during this period, let us know. Due to rapid publication and an extremely tight schedule, if comments are not received, we may publish the paper as it stands.

Open access

You are invited to opt for open access via our author pays publishing model. Payment of open access fees will enable your article to be made freely available via the Royal Society website as soon as it is ready for publication. For more information about open access publishing please visit our website at http://royalsocietypublishing.org/site/authors/open_access.xhtml.

The open access fee is £1,700 per article (plus VAT for authors within the EU). If you wish to opt for open access then please let us know as soon as possible.

Paper charges

Sincerely,

Proceedings B

Appendix A

COLEG GWYDDORAU'R AMGYLCHEDD A PHEIRIANNEG
COLLEGE OF ENVIRONMENTAL SCIENCES AND ENGINEERING

YSGOL GWYDDORAU EIGION
SCHOOL OF OCEAN SCIENCES

Registered charity number: 1141565

18th December 2019

Dear Prof. Cuthill,

We wish to thank you and the two anonymous reviewers for a positive response and valuable comments. Following the reviewer's comments, we have made several improvements to the manuscript. Here we summarise the main amendments, followed by detailed responses to each comment. Our responses are indicated in **blue text**. Line numbers refer to the tracked changes version of the revised manuscript, attached below.

Referee 1 makes the following main comments in their report:

1. "... my main reservation about the paper is that some of the content around the role of 3D data from TLS and SfM has already been covered in detail by previous reviews. This is particularly true in the context of forests, where TLS data are already revolutionising how we estimate biomass and study forest structure... So I guess in my mind the real novelty of what is being proposed in this review is the application of these technologies in new ecosystems (e.g., intertidal, marine etc.)"
We agree with the referee that the techniques discussed are seeing application in certain fields such as forest science, and that some of the information reviewed is available in other sources. Our aim is to introduce and review the techniques for a broad ecological readership that are less familiar with the available tools, and encourage their adoption in new areas of ecology where it has been far slower, rather than focus on applications in a particular sub-discipline. We have made improvements to the manuscript to further acknowledge and incorporate information from previous work including studies in forest ecosystems. Please see response to comment 1.02 below for specific changes.
2. "The other thing I think is worth mentioning in this context is that in addition to TLS and SfM from drones, the other emerging 3D technology that would be worth incorporating in this review is UAV LiDAR... So I think it would be worth incorporating this into the review as well as (i) discuss the pros and cons of these data when compared to SfM data from UAVs (e.g., in terms of cost, accuracy, coverage, information etc.) and (ii) discuss how the two can be fused..."
We accept that a number of other 3D mapping technologies are available and that UAV LiDAR is one such promising technique. We made a concerted effort to incorporate other techniques including UAV LiDAR and wrote a new 500 word section to review these technologies. However, on reflection, we felt that we couldn't do justice to a review of these other techniques and the complex considerations for their use within the page limits. We also felt that the inclusion of this new section distracted from the core aim, to encourage more widespread adoption of high-resolution 3D mapping

PRIFYSGOL BANGOR
YSGOL GWYDDORAU EIGION,
PORTHAETHWY, YNYS MÔN,
LL59 5AB, DU

FFÔN: +44 (01248) 351151

BANGOR UNIVERSITY
SCHOOL OF OCEAN SCIENCES,
MENAI BRIDGE, ANGLESEY
LL59 5AB, UK

TEL: +44 (01248) 351151

TIM D'URBAN JACKSON
SWYDDOG YMCHWIL/ RESEARCH OFFICER

RHIF UNIONGYRCHOL / DIRECT LINE: +44 (01248) 383967
EBOST / EMAIL: t.d.jackson@bangor.ac.uk

www.bangor.ac.uk

www.bangor.ac.uk/oceansciences

in ecology. UAV LiDAR is currently less accessible than terrestrial laser scanning and structure-from-motion photogrammetry for users with limited or no experience in the field. For these reasons we decided not to include additional content about UAV LiDAR as we think this technique deserves a thorough review in a separate piece of work.

Referee 2 makes the following main comments in their report:

1. "...important work must be done to place this review in context of a large body of work undertaken to understand ecological questions using increasingly high detail 3D ecosystem structure using techniques other than TLS and SfM."

We agree that more background content was needed to place the review of newly accessible techniques into the context of a wider body of work using increasingly advanced 3D remote sensing tools and data sources to address ecological questions. We have added to the manuscript to acknowledge passive and active satellite, and crewed aircraft-based remote sensing applications in ecology, and direct the reader to sources of further information about these data sources. Please see the response to comment 1.08 below.

2. "In addition, the authors should do more work to review or at least mention/cite specific challenges inherent to interpreting digitally derived 3D data of multi-layer/turbid medium ecosystems such as forests."

The challenges in interpreting 3D data of multi-layer ecosystems such as forests are indeed important to address, as are challenges specific to other systems. We have added new content to the "Barriers to wider uptake" section, which highlights these issues, citing specific challenges. Please see the response to comment 2.03 below.

Please find our detailed responses to each comment below and the revised text in each instance. We have fully engaged with the reviewers' comments and feel that the manuscript has improved as a result.

Kind regards,

Tim D'Urban Jackson
Gareth J. Williams
Guy Walker-Springett
Andrew J. Davies

Responses to specific comments

Referee 1

Comment 1.01

D'Urban Jackson and co-authors have done a very nice job of outlining how access to new ultra-high resolution 3D data has the potential to change how we study ecosystems in space and time. The review is well written and clearly structured, and the great effort put into the graphics really helps convey the potential of these new technologies. I'm sure that this paper would be of interest to a broad readership. Well done!

Response

Thank you for your encouraging comments

Comment 1.02

I've made some suggestions below that hopefully will help the authors refine their paper even further. As a terrestrial vegetation ecologist, my main reservation about the paper is that some of the content around the role of 3D data from TLS and SfM has already been covered in detail by previous reviews. This is particularly true in the context of forests, where TLS data are already revolutionising how we estimate biomass and study forest structure. In fact the Royal Society's Interface Focus journal recently had an entire special issue on exactly this (see here: <https://royalsocietypublishing.org/doi/full/10.1098/rsfs.2017.0052>). The same is true for SfM data, that are being increasingly used for biomass mapping (e.g., see here for one of the several reviews published on this in the last couple of years: <https://link.springer.com/article/10.1007/s40725-019-00094-3>). So I guess in my mind the real novelty of what is being proposed in this review is the application of these technologies in new ecosystems (e.g., intertidal, marine etc.)

Response

We agree with the referee that these techniques are seeing application in certain fields, and that some of the information is available in other sources. Our aim is to introduce and review the techniques for a broad ecological audience that are less familiar with the available tools, and encourage their adoption in new areas of ecology where it has been far slower, rather than focus on applications in a particular sub-discipline. We have made the following changes to further acknowledge previous work, including the two sources that the referee has recommended:

Line 195: More recently it has seen application in ecology [12], particularly in forest ecology where the below-canopy perspective complements airborne data collection. Applications include quantifying biomass, growth and 3D structure of forest vegetation [14,18–21], non-destructive estimation of above ground grass and mangrove biomass [22,23], assessing vegetation water content [24]...

Line 251: Ecological applications include modelling forest and vegetation structure and biomass [22,31,34,35]...

Line 367: Unique insights are already being generated, particularly in forest and coral reef ecosystems [48], whereas adoption has been slower in other systems such as intertidal habitats. Multiscale topography plays a critical structuring role in the intertidal zone by controlling environmental conditions and field time is constrained by tidal cycles, making rapid 3D mapping tools valuable to intertidal field ecologists.

Comment 1.03

The other thing I think is worth mentioning in this context is that in addition to TLS and SfM from drones, the other emerging 3D technology that would be worth incorporating in this review is UAV LiDAR. The technology has come a long way here and there are numerous groups around the world that are capturing extremely high resolution LiDAR from UAVs across scales that can't be covered using a TLS (just as one example of what is possible see here:

<https://www.sciencedirect.com/science/article/abs/pii/S0034425719303748>). So I think it would be worth incorporating this into the review as well as (i) discuss the pros and cons of these data when compared to SfM data from UAVs (e.g., in terms of cost, accuracy, coverage, information etc.) and (ii) discuss how the two can be fused (e.g.,

<https://royalsocietypublishing.org/doi/full/10.1098/rsfs.2017.0038>).

Response

We accept that a number of other 3D mapping technologies are available and that UAV LiDAR is one such promising technique. We made a concerted effort to incorporate other techniques including UAV LiDAR and wrote a new 500 word section to review these technologies. However, on reflection, we felt that we couldn't do justice to a review of these other techniques and the complex considerations for their use within the page limits. We also felt that the inclusion of this new section distracted from the core aim, to encourage more widespread adoption of high-resolution 3D mapping in ecology. UAV LiDAR is currently less accessible than terrestrial laser scanning and structure-from-motion photogrammetry for users with limited or no experience in the field. For these reasons we decided not to include additional content about UAV LiDAR as we think this technique deserves a thorough review in a separate piece of work.

Comment 1.04

L132-4: I'm not sure this statement is true. SfM has been rapidly adopted by a whole range of disciplines in ecology and conservation. Moreover, from a commercial standpoint many surveying companies actually prefer it to LiDAR as the software for processing the drone imagery has come a long way and the data can be acquired much more cheaply, using smaller drones and with less risk.

Response

We agree that the point needed refining. We have edited the statement to clarify the point that there is more uncertainty in structure-from-motion photogrammetry outputs than more established commercial survey techniques like terrestrial laser scanning.

Line 219: Commercial adoption of structure-from-motion is increasing as a low-cost, flexible survey tool, but questions remain over best practices for producing repeatable and high quality outputs.

Line 230: There is greater opportunity for error introduction with structure-from-motion compared to terrestrial laser scanning, and uncertainty in data outputs varies widely and unpredictably within [32] and among studies [33].

Comment 1.05

L141-2: one big difference between TLS (and LiDAR in general) and SfM that would be worth discussing is that LiDAR allows you to see 'through' vegetation and capture both the canopy and the

ground, whereas SfM is limited to the surface. This is actually a big limitation in structurally complex ecosystems like forests.

Response

We agree that we needed to better explain differences in the characteristics of terrestrial laser scanning and structure-from-motion data. We have made the following changes to address this comment.

Line 175: Terrestrial laser scanning is conducted from a set of discrete stations using a tripod-mounted instrument, collecting data radially from a low elevation (generally < 2 m). This results in a reduction in both point density and angle of incidence to the ground with increasing distance from the scanner, and sectors of missing data behind obstructions like trees. Regions with low point density are filled by merging data from multiple scanning stations (figure 2), introducing a low level of quantifiable error.

Line 182: Terrestrial laser scanning typically penetrates through fine-scale features like vegetation to record points on internal surfaces (e.g. branches) and the ground, as the independent laser pulses can travel through small gaps.

Line 238: Structure-from-motion generates more homogenous and comprehensive data coverage compared to terrestrial laser scanning in less time, because the camera is moved around the scene, often using an aerial platform. However, multiple images of a point on a feature are needed to calculate a position, so internal surfaces of complex features (e.g. branches of a dense bush or coral), shaded surfaces and moving features (e.g. blades of grass in the wind) are less likely to be captured or positioned accurately. Structure-from-motion tends to return a generalised outer surface of such features, lacking finer details.

Line 572: However, system-specific challenges remain in survey design, data processing and interpretation. With terrestrial laser scanning in complex environments, line-of-sight obstructions and moving vegetation combined with the spatial characteristics of the point cloud data generates challenges for interpretation and analysis [46,70,71]. While the moving vantage aspect of structure-from-motion data capture means more homogenous data coverage, repeatability of coral reef rugosity measurements were impacted by high habitat complexity, environmental conditions and variation in methods [72].

Comment 1.06

L195: when you say 'low cost' this is not really true for TLS. A good scanner (i.e., REIGL) costs >100K and is out of reach for most groups. By contrast a Phantom 4 drone is about £1K. This is a big distinction between the two technologies that's worth making in the paper.

Response

This is a valid point. We have made several changes to clarify that there is still a large difference in costs between the two techniques.

Line 135: rapidly deployable remote sensing tools

Line 189: Falling costs and improved portability have increased the accessibility of terrestrial laser scanning to a wide variety of users [14,16]. Custom built versions have lowered costs even further [16], although the equipment and software required is still expensive compared to structure-from-motion photogrammetry, and may be prohibitively so for some users.

Line 214: Structure-from-motion photogrammetry is a low-cost machine vision technique

Line 360: Terrestrial laser scanning and structure-from-motion photogrammetry offer rapid, detailed, continuous extent 3D mapping of ecosystems.

Line 517: Equipment, software and training costs can still be significant, especially for terrestrial laser scanning, with further costs incurred for maintenance and insurance. However, the multidisciplinary applications of the techniques mean many institutions will already have access to suitable equipment and software, or can gain access to shared resources. Structure-from-motion costs can be comparable to many other field techniques, especially if using a handheld camera and open source software.

Comment 1.07

L194-202: I would also mention the limitation of scale, particularly for TLS. Covering large area in ultra-high detail remains prohibitive in terms of time, cost and computing.

Response

We agree that this is an important point to make and have made the following changes:

Line 180: Data extent, resolution and coverage must be balanced with the survey time needed, especially in complex ecosystems like forests where many stations are required for comprehensive coverage of a large extent.

Line 536: Processing a basic structure-from-motion model can be achieved in a similar amount of time, but an accurate, detailed model typically takes a day or more to process depending on processing power and number of images.

Comment 1.08

L204: Arguably airborne LiDAR paved the way for high-resolution habitat suitability mapping. Worth mentioning this and discussing what TLS/SfM can give you that airborne LiDAR can't from this perspective. I would think this is likely to be highly dependent on the organism in question and there is likely to be a trade-off with spatial scale (i.e., airborne LiDAR can cover much larger areas).

Response

We agree that an overview of other remote sensing technologies from satellite and crewed aircraft platforms was needed to give a stronger background context to this work. We have added content to address this comment (Line 91-120).

Comment 1.09

L240-2: Good on paper, but in reality automating many of these processes is still a big bottleneck. I'm thinking for instance of the algorithm for segmenting trees from TLS that require a lot of manual tuning and pruning to get right. Worth mentioning.

Response

We agree that this point needed including and have made the following changes in response.

Line 584: Data processing still requires manual input at various stages, and automated workflows can be computationally demanding, especially for structure-from-motion. Various algorithms and software packages are being developed for 3D point cloud processing, including open source projects like CloudCompare [45]. After the initial processing stages required to generate a 3D model, further processing and analysis currently requires non-trivial technical skill or novel approaches specific to the task. As 3D methods become more common in ecology, an increase in demand and funding for user-friendly and powerful processing techniques, including packages for open-source platforms like Python and R, can be expected.

Comment 1.10

L250-3: I seem to remember similar results fro bird colonies (possibly here: <https://besjournals.onlinelibrary.wiley.com/doi/full/10.1111/2041-210X.12974>)

Response

This is an interesting paper, but as the authors us different techniques to those discussed here, we decided not to include it given the page limit for this work.

Comment 1.11

L330-346: I think an additional barrier you are overlooking here are the substantial costs associated with purchasing equipment, getting specialised training, obtaining permits/licences, insurance (for drones in particular) and running costs (batteries, software, part, maintenance etc).

Response

See response to comment 1.06

Comment 1.12

In terms of identifying hurdles to uptake one thing that would actually be really interesting to do (although not necessarily for this paper) would be to survey the research community and ask them what the barriers are (e.g., at the BES annual meeting).

Response

This would indeed be a very valuable exercise and something we will consider pursuing.

Referee 2

Comment 2.01

This is a well-written review and I believe fills a need in the literature. The insights are valuable and in my opinion this paper will further help bridge the gap between remote sensing scientists and ecologists.

Response

Thank you for the encouraging comments

Comment 2.02

That said, important work must be done to place this review in context of a large body of work undertaken to understand ecological questions using increasingly high detail 3D ecosystem structure using techniques other than TLS and SfM.

Response

See response to comment 1.08

Comment 2.03

In addition, the authors should do more work to review or at least mention/cite specific challenges inherent to interpreting digitally derived 3D data of multi-layer/turbid medium ecosystems such as forests.

Response

See response to comment 1.05

Comment 2.04

Line 41: Regarding the “recently emerged” comment: discrete return airborne scanning lidar capable of characterizing 3-D ecosystem structure at the meter scale was relatively mature 20 years ago. Terrestrial laser scanning research to characterize forest structure for ecological applications stretches back about 15 years. It is important to acknowledge this body of work here in the introductory portion of the paper. As is, the authors jump from mentioning “conventional methods” of ecological sampling to the “recently emerged” TLS and SfM methods. Airborne lidar is a critical stepping stone between the two in characterizing 3D ecosystem structure, and is still the only method available to characterize large swaths of land (i.e. tens to hundreds of hectares) in 3D with sub-meter grain.

Response

We acknowledge that the wording needed revising and have made the changes below. Please see response to comment 1.08 for our added content relating to background context of other remote sensing sources.

Line 33: We introduce two high-resolution remote sensing tools for rapid and accurate 3D mapping in ecology

Line 69: Disruptive remote sensing technologies to rapidly record detailed, spatially-referenced biological and physical information are now accessible to the field ecologist.

Line 145: **High-resolution remote sensing tools for spatial ecology**

Line 360: Terrestrial laser scanning and structure-from-motion photogrammetry offer rapid, detailed, continuous extent 3D mapping of ecosystems

Comment 2.05

Line 43: One can easily argue that TLS is not very accessible or powerful for characterizing forest habitats or topographically complex ecosystems (for two examples) across multiple scales, given the large number of scans and logistically challenging geocorrections it takes to characterize these structurally complex environments without significant data dropouts from occlusion effects. The authors are oversimplifying the details of TLS here. ALS needs to be mentioned here in my opinion. It occurs to me that the way to help focus the paper without confusing readers familiar with the large body of ecological work published on airborne lidar, would be to explain from the outset that the authors are focused on small extent, fine scale ecological questions (i.e. at the scale of 10⁰-10² square meters) where TLS and SfM are most effective.

Response

See responses to comments 1.05 and 1.08. We agree that the specific scales of interest to this work needed clarifying, and have made the following changes:

Line 73: This review considers tools able to capture three-dimensional (3D) ecosystem data at finer scales than can be achieved with more familiar remote sensing from satellites or crewed aircraft.

Line 122: Satellite and crewed aircraft remote sensing is irreplaceable for continuous mapping at up to global extents. However, the technique becomes logistically inappropriate when detailed information is required across smaller spatial extents (metres to hectares) or shorter time periods (hours to weeks) due to limits of data resolution, accuracy or cost. For 3D mapping at these scales, recent technological advances have led to the emergence of high-resolution (millimetre to centimetre), rapidly deployable remote sensing tools that include terrestrial laser scanning and structure-from-motion photogrammetry (figure 1)

Comment 2.06

Paragraph starting Line 55: Again there is a missing set of techniques that are not mentioned here. Airborne passive remote sensing data has long provided sub-meter information relevant to ecological study. Satellite passive remote sensing data also is highly relevant to high spatial resolution ecological processes (e.g. the Worldview series of sensors has gotten progressively finer in resolution over the past decade, to a point of submeter resolution panchromatic imagery, and just above one meter resolution multispectral imagery). And, as mentioned above airborne lidar has been providing sub-meter data for about 20 years. The authors should acknowledge these data types and make the case why they do not warrant discussion in an ecological context. Readers can be pointed toward recent applicable reviews such as Eitel et al. 2016 (<http://dx.doi.org/10.1016/j.rse.2016.08.018>) and Anderson and Gaston 2013 (<https://doi.org/10.1890/120150>). Nowak et al, 2019 (<https://doi.org/10.2478/eje-2018-0012>) also is a very recent and relevant paper relevant to the SfM portion of the current review.

Response

See response to comment 1.08. We have included the first two recommended sources. Upon consideration of the third source, we feel that the information it contains relating to structure-from-motion is covered by other, more specific work that we have cited.

Comment 2.07

Line 81: I would suggest using the term airborne laser scanning here, to show that it is essentially the same technology as terrestrial laser scanning, just mounted on a different platform.

Response

We have revised the terminology as suggested

Line 109: Airborne laser scanning has become a widely used tool for characterising 3D habitat structural complexity and exploring organism-habitat relationships

Line 160: Using the same principles as airborne laser scanning, terrestrial laser scanning is a high-precision ground-based survey technique used extensively in civil engineering

Comment 2.08

Line 84: Please be more specific about the wording here, as it is confusing. Rather than “calculating the positions of millions of reflected laser pulses emitted in all directions”, the instrument is only able to calculate the location of objects struck by the laser when a portion of the laser is then reflected in the exact same direction back to the TLS instrument.

Response

We agree that the wording needed revising. We have made the following changes

Line 162: It is an active remote sensing approach that builds an accurate model of the surroundings by emitting millions of laser pulses in different directions and analysing the reflected signals

Comment 2.09

Line 96: Yes this is a nice overview. The aforementioned review by Eitel et al. 2016 described many ecological applications of TLS and should be cited here for further reading.

Response

We thank the referee for bringing this valuable work to our attention and have included it as suggested.

Line 195: More recently it has seen application in ecology [12]

Comment 2.10

Paragraph starting Line 164: Yes this is a nice comparison, with good results. Thank you. I suggest that as this is a review, the authors overview some previous work done in ecosystems with more complex multi-layer structure where lower strata are often obscured from view of the drone. For example, the recent paper by Hillman et al. 2019 (10.3390/rs11182118) highlights some of the challenges in complex multi-layer ecosystems.

Response

We agree that more context to the results of this work was needed. We have included the paper suggested as well as other relevant studies in the following changes.

Line 354: Similar results are reported in other studies, with high agreement between structure-from-motion and terrestrial laser scanning at fine-scales of up to 1 m² [23,47] and centimetre-level accuracy at broad scales (hectares) [44,48]

Comment 2.11

Line 236: Very good to highlight water-based ecological applications. Exciting new areas of research for SfM

Response

Thank you

Comment 2.12

Line 240: While data acquisition may be rapid and relatively low-cost, it should be mentioned that the workflows required to process these data (particularly TLS in complex structure ecosystems) can be greatly time consuming and require non-trivial technical skill. (OK, in reading further I see that you cover this later).

Response

See response to comment 1.09

Comment 2.13

Line 245: yes, very good. This is the type of information/comparison that I was looking for earlier in the paper. It should occur earlier in my opinion in order to highlight the specific niche that this paper wishes to fill. However, the great amount of ecological work at increasingly finer scales using non-TLS and non-SfM approaches that has occurred since Turner et al.'s 2003 review warrants mention as well.

Response

See response to comment 1.08

Comment 2.14

Line 254: Very good point that as one wishes to examine increasingly finer spatial scales, the data must be evaluated at these finer scales to assess accuracy. This will continue to be a challenge, yet a nice problem to be able to try to solve now that the data exist at this scale.

Response

We have emphasised this point with the following change

Line 269: With sub-centimetre-resolution 3D data, georeferencing error can be a limiting factor for detection of fine-scale change in topography through time [30], and for estimating the accuracy of survey techniques [44], demanding positioning technology with sub-centimetre accuracy (e.g. Total Station).

Comment 2.15a

Line 317-329: Please do mention and cite here the challenges (albeit many opportunities exist as well_ associated with doing fine-scale ecology using these techniques in more structurally complex ecosystems. Citing work in forests for example should be done to make this paper more broadly applicable as a review.

Response

See response to comment 1.05

Comment 2.16

Line 351: to clarify and help differentiate your review from previous work please mention the fine scales you are focusing on, e.g. "...spatial ecology questions at the sub-square meter to tens-of square meters scale..."

Response

See response to comment 2.05

Comment 2.17

Line 357: please acknowledge previous work by acknowledging your review adds to the continuum of ecological work enabled via remote sensing at increasingly finer scales, e.g. "...spatial ecology is continuing to evolve, and high-resolution..."

Response

We have made the following change as suggested

Line 607: Digital ecosystem analysis and multiscale 3D spatial ecology is continuing to evolve, and high-resolution remote sensing techniques are becoming instrumental as part of the modern spatial ecologist's tool kit.

Comment 2.18

Figures: The figures are all relevant to the paper in my opinion, have useful captions, and have beautiful aesthetic appeal.

Response

Thank you very much